# The Short-Term Opening of Cyclosporin A-Independent Palmitate/Sr^2+^-Induced Pore Can Underlie Ion Efflux in the Oscillatory Mode of Functioning of Rat Liver Mitochondria

**DOI:** 10.3390/membranes12070667

**Published:** 2022-06-28

**Authors:** Natalia V. Belosludtseva, Lyubov L. Pavlik, Konstantin N. Belosludtsev, Nils-Erik L. Saris, Maria I. Shigaeva, Galina D. Mironova

**Affiliations:** 1Institute of Theoretical and Experimental Biophysics, Russian Academy of Sciences, Institutskaya 3, 142290 Pushchino, Russia; pavlikl@mail.ru (L.L.P.); bekonik@gmail.com (K.N.B.); shigaeva-marija@rambler.ru (M.I.S.); mironova40@mail.ru (G.D.M.); 2Department of Biochemistry, Cell Biology and Microbiology, Mari State University, pl. Lenina 1, 424001 Yoshkar-Ola, Russia; 3Department of Microbiology, Antimicrobials, Probiotics and Fermented Food, University of Helsinki, 00014 Helsinki, Finland; nils-erik.saris@helsinki.fi

**Keywords:** mitochondria, mitochondrial permeability transition, ion oscillations, phospholipase A_2_, palmitic acid, cyclosporin A, cyclosporin A-independent palmitate/Ca^2+^-induced permeability transition pore, lipid pore

## Abstract

Mitochondria are capable of synchronized oscillations in many variables, but the underlying mechanisms are still unclear. In this study, we demonstrated that rat liver mitochondria, when exposed to a pulse of Sr^2+^ ions in the presence of valinomycin (a potassium ionophore) and cyclosporin A (a specific inhibitor of the permeability transition pore complex) under hypotonia, showed prolonged oscillations in K^+^ and Sr^2+^ fluxes, membrane potential, pH, matrix volume, rates of oxygen consumption and H_2_O_2_ formation. The dynamic changes in the rate of H_2_O_2_ production were in a reciprocal relationship with the respiration rate and in a direct relationship with the mitochondrial membrane potential and other indicators studied. The pre-incubation of mitochondria with Ca^2+^(Sr^2+^)-dependent phospholipase A_2_ inhibitors considerably suppressed the accumulation of free fatty acids, including palmitic and stearic acids, and all spontaneous Sr^2+^-induced cyclic changes. These data suggest that the mechanism of ion efflux from mitochondria is related to the opening of short-living pores, which can be caused by the formation of complexes between Sr^2+^(Ca^2+^) and endogenous long-chain saturated fatty acids (mainly, palmitic acid) that accumulate due to the activation of phospholipase A_2_ by the ions. A possible role for transient palmitate/Ca^2+^(Sr^2+^)-induced pores in the maintenance of ion homeostasis and the prevention of calcium overload in mitochondria under pathophysiological conditions is discussed.

## 1. Introduction

Currently, there are numerous data indicating the involvement of mitochondria in the regulation of the intracellular dynamics of free Ca^2+^, including generation, distribution, and synchronization of Ca^2+^ waves in the cell cytoplasm [1,2,3,4]. Accumulating evidence suggests that the formation and propagation of “calcium waves” in cells is mediated by oscillations in ion fluxes in mitochondria [4,5]. In isolated mitochondria, a pulse addition of Ca^2+^ or its chemical analog Sr^2+^ can trigger complex dynamic modes of ion transport accompanied by fluctuations in transmembrane potential, respiration rate, volume of the mitochondrial matrix, and ATP production [6,7,8,9,10,11,12].

As early as 50 years ago, it was shown that mitochondria could generate ion oscillations with a wide variety of periods and waveforms. The history of the study of oscillatory processes in mitochondria begins with the works of B. Chance, L. Packer, H. Lardy, B. Pressman, and others [13]. The main stimulus for the development of mitochondrial oscillations was found to be the transport of the divalent metal ions Ca^2+^ or Sr^2+^ across the inner mitochondrial membrane. The co-addition of the potassium ionophore valinomycin to mitochondria promoted long-term fluctuations in ion fluxes across the mitochondrial membrane. Studies by A. Gyulkhandanyan, Yu. Evtodienko, and others demonstrated sustained Sr^2+^/valinomicin-induced oscillations in the Sr^2+^, K^+^ and H^+^ fluxes, oxygen consumption, and light scattering of suspensions of isolated rat liver mitochondria under hypotonic conditions [5,6,11].

Mitochondria are now recognized to contain several systems involved in Ca^2+^ transport: the mitochondrial Ca^2+^ uniporter complex (MCUC), the rapid mode of Ca^2+^ uptake (RaM), the leucine zipper and EF-hand-containing transmembrane protein 1 (Letm1), mitochondrial ryanodine receptor type 1, uncoupling proteins (UCPs), Na^+^/Li^+^/Ca^2+^ exchanger (NCLX), the Ca^2+^/H^+^ exchanger, and the mitochondrial permeability transition pore (mPTP) [14,15,16]. Among them, MCUC is believed to be the predominant calcium import system, which plays a crucial role in controlling ion uptake by mitochondria under pathophysiological states [12,16,17]. It was found that mitochondrial influx of Ca^2+^(Sr^2+^) upon Ca^2+^(Sr^2+^)-induced ion oscillations occurred primarily via MCUC [6,7,8,9,10,11,12]. As for the mechanism of ion efflux from mitochondria in oscillatory modes, this issue is still not clear. Some studies have shown that the opening of the mPTP, a multiprotein mega-channel complex at outer and inner membrane contact sites, is not related to the release of ions from mitochondria in Ca^2+^(Sr^2+^)-induced ion oscillations, since the selective mPTP blocker cyclosporin A (CsA) did not affect the cycling of ions across mitochondrial membranes [12,18,19,20,21]. In addition, the loading of mitochondria with Sr^2+^ ions, which are unable to induce mPTP opening, not only did not prevent but even favored continuous ion oscillations in mitochondria [8,10,11]. In this regard, the participation of the mPTP in Ca^2+^(Sr^2+^)-induced self-oscillations in ion fluxes has been questioned.

Earlier, we found that one of the pathways of ion transport can be through the formation of lipid pores in mitochondrial and artificial (liposomes and black lipid) membranes by the mechanism of chemotropic phase transition [22,23,24]. The mechanism can be realized due to the capacity of long-chain saturated fatty acids to bind Ca^2+^ with high affinity [25]. Unlike the mPTP, the CsA-independent palmitate/Ca^2+^-induced permeability transition pore (PA-mPT pore) can be induced by lower concentrations of Ca^2+^ or Sr^2+^ ions in the mitochondrial matrix under conditions close to physiological ones [26,27,28,29]. Furthermore, one of the main features of the PA-mPT pore is its ability to spontaneously close, with a rapid restoration of membrane integrity [26,29,30]. As is already known, the ability of lipid pores to rapidly heal, if their diameter does not exceed a certain value, is an intrinsic property [31]. It has been observed that induction of the PA-mPT pore can lead to the transient permeabilization of the mitochondrial membrane, but the mitochondria remain functionally active [26,29,30]. We suggest that the intermittent opening/closure of the PA-mPT pore underlies the so-called “flickering” mode of permeability transition, which is reversible and appears to have a role in maintaining mitochondrial calcium homeostasis by providing the organelles with an emergency pathway for the rapid outflow of Ca^2+^ ions [30].

The aim of this work was to reveal whether PA-mPT pore formation is involved in the mechanism of ion efflux upon ion fluctuations triggered by Sr^2+^ in the presence of valinomycin in rat liver mitochondria. Using a number of inhibitors of Ca^2+^(Sr^2+^)-dependent phospholipase A_2_, we found that a decrease in the content of free fatty acids, including palmitic and stearic acids, suppressed continuous Sr^2+^-induced cyclic changes in ion fluxes and the main functions of mitochondria.

## 2. Materials and Methods

### 2.1. Materials

SrCl_2_ and tetraphenylphosphonium chloride were purchased from Merck KGaA (Darmstadt, Germany); arachidonyl trifluoromethyl ketone (AACOCF_3_), 4-(4-Octadecylphenyl)-4-oxobutenoic acid (OBAA), and palmitoyl trifluoromethyl ketone (PACOCF_3_) were obtained from Tocris BioScience (Bristol, UK); aristolochic acid I (AA), 6E-bromoenol lactone (BEL), trifluoperazine dihydrochloride (TFP), 4′-bromophenacyl bromide (BrB), valinomycin (Val), CsA, and other chemicals were purchased from Sigma-Aldrich (St. Louis, MO, USA). The stock solutions of AACOCF_3_, PACOCF_3_, and OBAA were prepared in dimethylsulfoxide (DMSO). The stock solutions of AA, BEL, TFP, BrB, Val, and CsA were prepared in 95% ethanol. In the control experiments, an equivalent volume of the solvent was used. The final concentrations of DMSO and ethanol in the incubation medium did not exceed 0.1% (volume).

### 2.2. Animals

Experiments were performed on Wistar male rats weighting 200–250 g. The animals were housed under standard conditions at a room temperature of 18–22 °C, relative humidity 60–70%, and regular 12 h light-dark cycles and received commercial pellets and water ad libitum. All assays were carried out in accordance with Protocol No. 19/2020 of 18.02.2020 of the Ethics Committees at the Institute of Theoretical and Experimental Biophysics of the Russian Academy of Sciences and the University of Helsinki.

### 2.3. Isolation of Rat Liver Mitochondria

Mitochondria were isolated from the liver of mature male Wistar rats (220–250 g) by a standard differential centrifugation technique using a Sigma 3-16K centrifuge (Sigma-Aldrich, St. Louis, MO, USA) [27]. The livers were cooled in saline, pressed through a plate, and homogenized with a Potter-type glass homogenizer. The medium contained 210 mM mannitol, 70 mM sucrose, 10 mM HEPES/KOH (pH 7.4), and 1 mM EDTA. The homogenate was centrifuged at 700× *g* (10 min), and mitochondria were sedimented for 15 min at 7000× *g*. The mitochondrial pellet was resuspended in a washing medium containing 210 mM mannitol, 70 mM sucrose, 10 mM HEPES/KOH (pH 7.4), and 0.1 mM EGTA and centrifuged for 15 min at 7000× *g*. The pellet was resuspended in the washing medium (0.1 mL/g of the liver tissue) and stored on ice. The concentration of mitochondrial protein was determined by the Lowry method [32]. The resulting suspension of rat liver mitochondria contained 70–80 mg of protein per mL.

### 2.4. Estimation of the Functional Parameters of Mitochondria

The mitochondrial membrane potential (ΔΨ_m_) was estimated by the distribution of the lipophilic cation tetraphenylphosphonium (TPP^+^) (1.5 µM), which was measured with a TPP^+^-sensitive electrode (Nico-Analyt, Moscow, Russia). Changes in the concentration of TPP^+^ in the incubation medium were inversely proportional to changes in the membrane potential [33]. The concentrations of Sr^2+^ and K^+^ ions in the incubation medium were determined with Sr^2+^- and K^+^-selective electrodes (Nico-Analyt, Moscow, Russia). Changes in the medium pH were registered by a pH microelectrode InLab Micro (Metler Toledo, Switzerland). Changes in the concentrations of TPP^+^, K^+^, H^+^, and Sr^2+^ were recorded simultaneously in a 1 mL temperature-controlled cell with constant stirring at 26 °C using an original multichannel electrometrical system Record 4 (Pushchino, Russia), as described previously [12]. The ion-selective electrodes were calibrated at the beginning of each experiment.

The swelling of mitochondria (0.4 mg/mL) was measured as a decrease in absorbance at 540 nm (*A*_540_) in a stirred cuvette at room temperature (24 °C) using a USB-2000 spectroscopy fiber-optic system (Ocean Optics, Dunedin, FL, USA) [26].

The rate of oxygen consumption by mitochondria was measured polarographically using an Oxygraph-2k (O2k) high-resolution respirometer equipped with DatLab software (Oroboros Instruments, Innsbruck, Austria), as described previously [34]. Temperature was maintained at 25 °C under continuous stirring at 600 rpm. The experiments were carried out with oxygen concentrations in the range of 220–50 µM O_2_. In additional control experiments, the respiratory control ratio (RCR) was measured under conventional conditions [34]. The RCR of isolated mitochondria was in the region of 5–6 when using 5 mM potassium succinate as a respiration substrate.

The rate of H_2_O_2_ production in mitochondria was determined using the fluorescent dye Amplex red (AR), as described previously [34]. The fluorescence of resorufin, an oxidized product of AR (excitation/emission at 563/587 nm) was measured using a CARY fluorimeter (Varian Inc., Palo Alto, CA, USA) at 36 °C under continuous stirring. The medium was supplemented with 10 μM AR and 1 U/mL of horseradish peroxidase; then, 0.1 mg/mL of the mitochondrial protein was added. The level of H_2_O_2_ was determined from the calibration curve. The concentration of a standard H_2_O_2_ solution was calculated using the molar absorption coefficient E_240_ = 43.6 M^−1^·cm^−1^. The rate of H_2_O_2_ generation was calculated based on the first derivative function of the H_2_O_2_ concentration vs. time and expressed as pmol H_2_O_2_/s·mL.

The incubation medium contained 20 mM sucrose, 1 mM KCl, 1 μM tetraphenylphosphonium chloride (TPP^+^), 1 μM cyclosporin A (CsA), 1 μM rotenone, 5 mM succinic acid, and 12.5 mM Tris (pH 7.3). SrCl_2_ and valinomycin were added to mitochondria 1 min after the start of incubation. The inhibitors of phospholipase A_2_ were added to the mitochondria 1 min before the addition of Sr^2+^. Stock solutions of the inhibitors were prepared in ethanol. The final concentration of ethanol in the incubation medium was <0.1 volume percent.

### 2.5. Determination of Free Fatty Acids upon Mitochondrial Oscillations by Gas Chromatography

The levels of free fatty acids (FFAs) in rat liver mitochondria were determined by gas chromatography using a Pye-Unicam 304 gas chromatograph (Pye Unicam Ltd., Cambridge, UK) equipped with a (1 m × 2 mm ID) glass column packed with Porapak Q, 80–100 mesh (Fluka Chemie AG, Buchs, Switzerland), in accordance with the conventional technique [35]. The content of main FFAs was analyzed before and 12 min after the addition of Sr^2+^ and valinomycin to the mitochondria in the absence (0.1% DMSO) or presence of 25 μM aristolochic acid (ArA), a phospholipase A_2_ inhibitor. The inhibitor of PLA_2_ ArA or 0.1% DMSO were added to mitochondrial suspensions 1 min before the addition of Sr^2+^ and valinomycin. The oscillatory mode of mitochondrial functioning was confirmed spectrophotometrically by reversible changes in optical density at 540 nm. The mitochondrial suspensions were fixed with a mixture of chloroform–methanol in the volume ratio 2:1. All samples were supplemented with 300 µg heptadecanoic acid (C17:0), which was used as an external standard. Chloroform and water–methanol layers were separated by centrifugation at 10,000× *g* for 10 min at 4 °C. The lower layer of the extract was collected, and a specific amount of anhydrous sodium sulfate was added for dehydration. The dehydrated extract liquor was dried under a gentle stream of nitrogen gas at room temperature. The obtained fractions were methylated, as described previously [35]. Methyl ethers were extracted by four volumes of hexane, and an aliquot was injected into the chromatograph. The injector and detector were kept at 200 and 250 °C, respectively. The carrier gas was nitrogen at a flow rate of 10 mL/min. Values were expressed as µg FFA per mg mitochondrial protein and represent the means of four samples.

### 2.6. Detection of the Group IV Cytosolic Phospholipase A_2_ in Isolated Rat Liver Mitochondria by Immunoelectron Microscopy

Immunoelectron microscopy of isolated rat liver mitochondria was performed to determine the distribution pattern of Ca^2+^(Sr^2+^)-dependent cytosolic phospholipase A_2_ (cPLA_2_) in the organelles. The immunochemical analysis was performed using the commercial group IV cPLA_2_ antibody (cat. no. sc-438, Santa Cruz Biotechnology, Dallas, TX, USA) and 10 nm colloidal gold-labeled antibody (cat. no. G7402, Sigma-Aldrich, St. Louis, MO, USA) as primary and secondary antibodies, correspondingly. The suspensions of rat liver mitochondria (1 mg/mL) were fixed at room temperature for 4 h in 4% paraformaldehyde and 0.5% glutaraldehyde solution in 0.1 M PBS buffer (pH 7.4). After washing with the buffer, the mitochondrial suspensions were postfixed for 2 h with a 1% solution of osmic acid in PBS, dehydrated in alcohols of increasing concentrations, and enclosed in Epon-812. The treatment of mitochondria with the antibodies was carried out in a dampening chamber on the grids with ultrathin sections, which were prepared on an ultramicrotome Leica EM UC6 (Wetzlar, Germany). To get rid of resin and osmium, the slices were incubated in solutions of periodic acid and sodium periodate for 3 min, with further washing with 0.1 M PBS buffer. The nonspecific labeling of proteins was blocked by the superblock reagent (Pierse, Rockford, IL, USA). All further procedures were performed using PSB buffer supplemented with 0.1% glycine and 0.1% Triton X-100. The cPLA_2_ antibody was used in a 1:50 dilution. The procedure was performed at 4 °C during the night. After thorough washing, the ultrathin sections were treated with the secondary antibody conjugated with 10 nm colloidal gold microparticles (dilution 1:20) for 1 h at room temperature, washed and stained with lead citrate and uranyl acetate. In control experiments, 0.1 M PBS buffer was used in place of the primary antibody (in the presence of the secondary antibody) as a negative control. The slices were viewed using a Tesla BS-500 electron microscope (Brno, Czech Republic) and scanned using an Epson V700 scanner (Epson, Long Beach, CA, USA). About 30 electron microscopic preparations were examined.

### 2.7. Data Processing

The data are expressed as the means ± standard derivation (m  ±  SD). Characteristic curves typical for each of the independent experiments (*n* = 7–10) are presented. Statistical analysis of the data was carried out using GraphPad Prism version 6.0 software for Windows (San Diego, CA, USA). Normality of the sample distributions was verified using the Shapiro–Wilk test before using parametric analyses. Repeated-measures analysis of variance (ANOVA) was used with post hoc Bonferroni tests to compare FFA levels. The differences were considered statistically significant at *p* < 0.05.

## 3. Results

### 3.1. Generation of Spontaneous Oscillations in Ion Fluxes and Respiration Rate of Rat Liver Mitochondria

Figure 1 shows that a single addition of Sr^2+^ (45 nmol/mg protein) to rat liver mitochondria energized by succinate and incubated in hypotonic medium triggered short-term synchronous oscillations (two oscillating waves) of fluxes of Sr^2+^, K^+^, and TPP^+^ ions across the mitochondrial membrane, as well as reversible changes in the rate of O_2_ consumption. As one can see, the mitochondrial membrane potential (ΔΨ_m_) estimated by the distribution of the molecular sensor TPP^+^ and the mitochondrial respiration rate recovered rapidly, and the added Sr^2+^ was eventually accumulated by the mitochondria. At the same time, the influx of K^+^ ions into the mitochondria was significantly slowed down. Under the resting-state conditions, the rate of transport of K^+^ across the mitochondrial membrane is known to be much lower than that of Ca^2+^ or Sr^2+^ [36,37]. One can suppose that the low rate of potassium transport limits the propagation of oscillating waves and is the main reason for the rapid damping of the Sr^2+^-induced spontaneous oscillations in ion fluxes and ΔΨ_m_.

As seen from Figure 2, the co-addition of the potassium ionophore valinomycin to the mitochondrial suspension increased the number of spontaneous Sr^2+^-induced oscillation waves of the fluxes of Sr^2+^ and K^+^ ions, as well as ΔΨ_m_. The long-term (within 30 min) reversible oscillations in ion fluxes and ΔΨ_m_ were induced by the addition of Sr^2+^ ions at concentrations of 35–50 nmol/mg protein and small amounts of valinomycin (2 ng/mg protein); in this case, three to five “cycles” of spontaneous ion oscillations in mitochondria could be observed (Figure 2).

Along with a temporary dissipation of ΔΨ_m_, the influx of Sr^2+^ and K^+^ ions into mitochondria was also accompanied by a synchronous decline in the pH_out_ of the medium (alkalization of the mitochondrial matrix) (Appendix A). The decrease in the electrochemical transmembrane potential was enhanced with each new wave of the oscillations, which ultimately impaired the ability of mitochondria to accumulate Sr^2+^, K^+^, and H^+^ ions. Our experiments suggest that the number of oscillation waves depends on the degree of coupling of isolated mitochondria and that the higher this is, the greater number of waves is registered.

Figure 3 shows that the cyclic transport of K^+^ ions across the mitochondrial membrane was accompanied by periodic changes in the volume of the mitochondrial matrix, as determined by the light scattering technique. One can see that the cycles of the low-amplitude swelling of mitochondria and their subsequent spontaneous contraction correspond to the influx and efflux of K^+^ ions, respectively. Following the last oscillation wave, irreversible mitochondrial swelling was observed.

Parallel studies of the dynamic changes in the rates of mitochondrial oxygen consumption and H_2_O_2_ generation under the experimental conditions demonstrated that these indicators also changed in an oscillatory mode (Figure 4A). One can see that when mitochondrial respiration decreased, that is, the degree of reduction of the electron transport chain complexes was higher, the rate of H_2_O_2_ production was maximal. Thus, there was an inverse relationship between the dynamic changes in mitochondrial respiration and H_2_O_2_ production. It should also be noted that the temporary decrease in ΔΨ_m_ corresponded to a synchronous drop in the rate of H_2_O_2_ production (Figure 4B). On the contrary, when the mitochondrial membrane potential returned to a high level, the rate of H_2_O_2_ formation also became maximal, which indicates a direct relationship between the dynamic changes in ΔΨ_m_ and H_2_O_2_ production. These synchronous periodic changes in mitochondrial function could persist for 20–30 min.

The specific blocker of MCUC ruthenium red (1 µM) suppresses the formation of the next oscillation wave of Sr^2+^ flux (Appendix A) and the other functional parameters of mitochondria. In contrast, the inhibitor of cyclophilin D, a key regulator of mPTP opening, cyclosporin A (CsA), did not affect mitochondrial oscillations, and it was added to the incubation medium in all the above experiments to exclude mPTP-related mitochondrial damage.

### 3.2. Accumulation of Free Fatty Acids and Effects of Inhibitors of PLA_2_ on the Long-Term Oscillatory Mode of Mitochondrial Functioning

It is known that, under hypotonic conditions, the activity of phospholipases A_2_ (PLA_2_) in mitochondria is enhanced [38,39]. The influx of Sr^2+^ in the mitochondrial matrix can also contribute to the activation of PLA_2_ and the accumulation of free fatty acids (FFAs) [39].

Our data showed that after the addition of Sr^2+^ ions and valinomycin, the level of total FFAs in rat liver mitochondria increased 1.7 times (Figure 5 and Appendix A). Among saturated long-chain fatty acids, palmitic (C16:0) and stearic (C18:0) acids accounted for the largest proportion; their total contents more than doubled. The addition of the inhibitor of Ca^2+^(Sr^2+^)-dependent PLA_2_ aristolochic acid (25 µM) before the start of long-term ion oscillations in mitochondria significantly blocked the accumulation of FFAs, including palmitic and stearic acids (Figure 5).

As has been found earlier, free palmitic and stearic acids in complexes with Sr^2+^(Ca^2+^) ions can induce the opening of the PA-mPT short-lived pores that are insensitive to CsA and other blockers of the classical mPTP [26,29,30]. To elucidate whether the PA-mPT pore is involved in the ion efflux pathway under long-term Sr^2+^-induced ion oscillations, we investigated the effect of several inhibitors of Ca^2+^(Sr^2+^)-dependent and Ca^2+^(Sr^2+^)-independent PLA_2_ on oscillation propagation.

Figure 6 shows the synchronous changes in the ion fluxes of mitochondria pre-incubated with the Ca^2+^(Sr^2+^)-dependent PLA_2_ inhibitor aristolochic acid (ArA). One can see that the incubation of mitochondria with ArA at a concentration of 25 µM eliminated Sr^2+^-induced cyclic changes in Sr^2+^ and K^+^ ion transport, ΔΨ_m_, mitochondrial matrix volume, as well as the rates of mitochondrial respiration and H_2_O_2_ production.

Similar effects were observed with other inhibitors of Ca^2+^(Sr^2+^)-dependent PLA_2_, namely, 15 µM arachidonyl trifluoromethyl ketone (AACOCF_3_), 1 µM 4-(4-Octadecylphenyl)-4-oxobutenoic acid (OBAA), 10 µM trifluoperazine dihydrochloride (TFP), 40 µM 4′-bromophenacyl bromide (BrB), and 15 µM bromoenol lactone (BEL) (Table 1, Appendix A). At the same time, the inhibitor of Ca^2+^-independent PLA_2_ palmityl trifluoromethyl ketone (PACOCF_3_, 20 µM) was ineffective (Appendix A).

### 3.3. Ultrastructural Localization of the Group IV Ca^2+^(Sr^2+^)-Dependent Cytosolic Phospholipase A_2_ in Isolated Rat Liver Mitochondria

Recent data suggest that only Ca^2+^(Sr^2+^)-dependent cytosolic phospholipase A_2_β3 (group IVB PLA_2_; encoded by *PLA2G4B*) is an endogenous protein and constitutively associated with mitochondria in liver tissue [40]. However, the exact localization and distribution of this type of PLA_2_ in the organelles has not yet been determined.

Using specific anti-cPLA_2_ and 10 nm colloidal gold-labeled secondary antibodies, we performed an immunoelectron microscopic study to determine the localization and distribution pattern of the group IV Ca^2+^(Sr^2+^)-dependent cPLA_2_ in isolated rat liver mitochondria (Figure 7).

As can be seen, electron-dense (black) gold labels were predominantly located on the outer and crista membranes of mitochondria. It should be noted that no colloidal gold labels were observed in the control experiments, when the primary antibody was replaced with 0.1 M PBS buffer (Appendix A).

## 4. Discussion

In the current study, we provide evidence relating to the oscillatory mode of mitochondrial function when a single pulse addition of Sr^2+^ (35–50 nmol/mg protein) to rat liver mitochondria in a hypotonic medium initiates the cycling of ions across the inner mitochondrial membrane. The triggering of long-term oscillations in ion fluxes required an increase in the permeability of the mitochondrial membrane to potassium ions. This can be explained by the fact that, under the resting-state conditions, the rate of the transport of K^+^ ions across the mitochondrial membrane is one order of magnitude lower than that of Ca^2+^ or Sr^2+^ [36,37,41]. Therefore, potassium influx can limit the next oscillation wave. The addition of small amounts of the highly specific potassium ionophore valinomycin resulted in prolonged oscillations in the following parameters of mitochondria: (1) membrane potential; (2) fluxes of Sr^2+^, K^+^, and H^+^ ions across the membrane; (3) the volume of the mitochondrial matrix; (4) the rate of mitochondrial respiration; and (5) the rate of H_2_O_2_ production. Additional experiments showed that the specific inhibitor of the mPTP opening CsA did not affect ion cycling in the mitochondria, which indicates that the efflux of ions in this mode of mitochondrial function could not be related to the opening and closure of the mPTP pore. Furthermore, strontium ions were used instead of Ca^2+^, as they are not able to induce mPTP pore opening [42,43]. Another feature of the Sr^2+^/valinomycin-induced mitochondrial oscillations is their strong dependence on the state of strontium transport. It is known that strontium ions enter mitochondria via the same pathway as Ca^2+^, i.e., MCUC [2,37]. The specific MCUC inhibitor RR rapidly stopped all the Sr^2+^/valinomycin-induced cyclic changes in ion fluxes in the mitochondria.

The fact that valinomycin can promote periodic changes in the movement of K^+^ and H^+^ ions across the mitochondrial membrane and in mitochondrial matrix volume has been found by many authors, starting with the classical works of B. Chance, R. Packer, and others [6,13]. However, the molecular mechanism of K^+^ release in exchange for H^+^ in mitochondria is as yet unclear [41]. With regard to the release pathway of Sr^2+^(Ca^2+^), K^+^, and other ions from mitochondria in the oscillatory mode, we suggest that it is mediated by the opening of mitochondrial palmitate/Ca^2+^-induced short-lived lipid pores. Our study showed that the accumulation of free Sr^2+^ in the mitochondrial matrix was accompanied by the hydrolysis of membrane phospholipids and the accumulation of endogenous FFAs, while the level of palmitic and stearic acids increased more than two times. It is known that mitochondrial PLA_2_s (the Ca^2+^-dependent cPLA_2_β and Ca^2+^-independent iPLA_2_γ, iPLA_2_β) can hydrolyze both sn-1 and sn-2 fatty acyl groups of phospholipids [40,44,45,46], with saturated fatty acids being predominantly at the sn-1 position [47]. Current data indicate that the group IV Ca^2+^-dependent cPLA_2_β3 (group IVB PLA2, encoded by *PLA2G4B*) displays PLA_1_, PLA_2_, and more powerful lysophospholipase activities and is constitutively associated with mitochondria [40]. Our results suggest that the Ca^2+^-dependent cPLA_2_ is localized mainly on the crista membranes of rat liver mitochondria. The accumulation of saturated long-chain FFAs in liver mitochondria incubated with divalent cations was confirmed previously [38,46], and their participation in Ca^2+^/H^+^ and K^+^/H^+^ exchange in the organelles was proposed [48]. Recent data also suggest that fatty acids in the presence of calcium ions can induce self-sustaining fluctuations in transmembrane voltage and current in biomimetic membranes without proteins [49,50].

As has been found earlier, palmitic and stearic acids can bind with Ca^2+^ ions with high affinity [25], and the formation of complexes of these fatty acids with Ca^2+^ in the membrane leads to the opening of short-lived Pal-mPT pores by the mechanism of the chemotropic phase transition in the lipid bilayer, as described previously for artificial membranes (liposomes and black lipid membranes) [22,23,24,25], erythrocytes [51], and mitochondria [21,22]. The formation of the non-selective lipid pore ensures the transport of ions along the gradient of their concentrations: the release of Sr^2+^ and K^+^ from mitochondria and the entry of H^+^ into the mitochondrial matrix. This promotes transient mitochondrial depolarization, and shrinking mitochondria become more condensed and their optical density is increased. Due to the ability of lipid pores to close spontaneously, the integrity and low permeability of the mitochondrial membrane to ions can be restored [21,22,26,29]. Pore closure contributes to the recovery of ΔΨ_m_, the main driving force of ion transport in mitochondria, K^+^ re-uptake by mitochondria (in complex with valinomycin), and swelling of the organelles (optical density is decreased). Under the conditions of recovered ΔΨ_m_, the mitochondria are able to re-uptake the released Sr^2+^ ions via MCUC, and the cycle can be repeated.

The experiments on pre-incubation of mitochondria with Ca^2+^(Sr^2+^)-dependent PLA_2_ inhibitors showed that the suppression of the enzyme activity led to a decline in the content of FFAs in the mitochondria in response to a Sr^2+^ pulse and the prevention of the second and subsequent spontaneous oscillation waves of the membrane potential, along with declines in the cycling of Sr^2+^ and K^+^ ions across the membrane, matrix volume, as well as spontaneous periodic changes in the rates of mitochondrial respiration and H_2_O_2_ generation. These results are consistent with our previous data showing that palmitic acid can re-start the CsA-insensitive Sr^2+^-induced release of Sr^2+^ after it has been abolished by the Ca^2+^-dependent PLA_2_ inhibitor AACOCF_3_ [12]. Therefore, it becomes clear that, to produce Sr^2+^-induced spontaneous oscillations in mitochondria, several factors are needed. One of the factors is a sharp increase in free Sr^2+^(Ca^2+^) in the mitochondrial matrix. It should be noted that, with slow continuous infusion of the same amount of strontium into a suspension of mitochondria, ion oscillations do not occur [10]. It may be that this is due to the binding of Sr^2+^ in the mitochondrial matrix and a decrease in the concentration of free ions required for the induction of Pal-mPT pores in the mitochondrial membrane. Another factor for triggering spontaneous oscillations in ion fluxes is an increase in the activity of mitochondrial PLA_2_ and accumulation of endogenous FFAs. The next factor for the occurrence of prolonged spontaneous oscillations in mitochondrial parameters is the maintenance of respiratory chain activity and membrane potential. As can be seen from Figure 2, a decrease in ΔΨ_m_ in each subsequent cycle attenuates ion oscillations; Sr^2 +^ is accumulated by mitochondria at a lower rate, and the intramitochondrial concentration of the free ion may be insufficient to induce the opening of Pal-mPT pores. Most probably, under physiological conditions, mitochondrial oscillations exist for a longer time.

Accumulating evidence shows that the pulse mode of changes in the concentrations of ions in the cytoplasm and mitochondria has physiological importance [4,5,52,53]. This way of transmitting the signal within the cell is faster than simple diffusion and provides the spatiotemporal regulation of Ca^2+^-dependent cell functions, since the encoding of a calcium signal can be realized via the modulation of the amplitude and frequency of oscillations in ion concentration. Synchronous Ca^2+^ oscillations in mitochondria are involved in the propagation of intracellular calcium waves that regulate multiple signaling cascades in the cytoplasm of both excitable and non-excitable cells [52,53,54].

A transient change in membrane permeability to calcium ions due to the formation of short-lived pores in the mitochondrial membrane may regulate intracellular calcium homeostasis and the volume of mitochondria, thereby protecting them from calcium overload and osmotic shock. As is known, mitochondrial calcium overload is the major cause of the formation of the mPTP, the collapse of mitochondrial membrane potential, and, ultimately, the induction of cell death [55,56]. The Pal-mPT pore may be one of the systems for a rapid release of calcium ions and the maintenance of mitochondrial function under conditions accompanied by a sharp increase in [Ca^2+^] in the matrix. The futile cycles of the ion mediated by MCUC and Pal-mPT pore functioning in mitochondria may lead to a mild uncoupling accompanied by a significant decrease in ROS production. As shown in this work, there is a direct relationship between the dynamic changes in the rate of H_2_O_2_ production by mitochondria and the value of the mitochondrial membrane potential (Figure 4). Earlier, a similar correlation has been shown only in static models using a low concentration of the uncouplers of oxidative phosphorylation and blockers of mitochondrial respiration [57,58]. The data confirm the idea that cells may be protected from oxidative stress and over-reduction of respiratory chain complexes by a mild but persistent decrease in the membrane potential of mitochondria [58,59]. It is known that membrane phospholipids are the primary targets of ROS. Periodic influx and efflux of Ca^2+^ during Ca^2+^-induced oscillations in mitochondria may provide hydrolysis of oxidized phospholipids by Ca^2+^-dependent PLA_2_ in the phase of Ca^2+^ accumulation in the matrix, followed by a decrease in the activity of phospholipase A_2_ after the ion release. This suggestion is supported by the data on periodic changes in the content of lysophospholipids in mitochondria during Sr^2+^-dependent oscillations [60]. Therefore, the oscillatory mode of mitochondrial function may contribute to the protection of cells and mitochondria from oxidative stress and calcium overload, which is associated with the development of many pathologies, such as cardiovascular diseases, neurological disorders, diabetes, obesity, cancer, and others [4,5,16,61].

## 5. Conclusions

In this work, it was found that, during prolonged ion oscillations in response to a Sr^2+^ pulse, the efflux of ions from mitochondria down their concentration gradients can be mediated by the opening of the short-living pore induced by fatty acids (mainly, palmitic and stearic acids) and Sr^2+^. The data obtained allow us to suggest that the transient palmitate/Sr^2+^(Ca^2+^)-induced pore is a tool for the rapid release of calcium ions from mitochondria, which may be involved in preventing mitochondrial calcium overload and regulating cell ion homeostasis. Studies of the Pal/Ca^2+^(Sr^2+^)-induced lipid pore hold promise for understanding the mechanism of the oscillatory mode of mitochondrial function under physiological and pathological conditions.

## Figures and Tables

**Figure 1 membranes-12-00667-f001:**
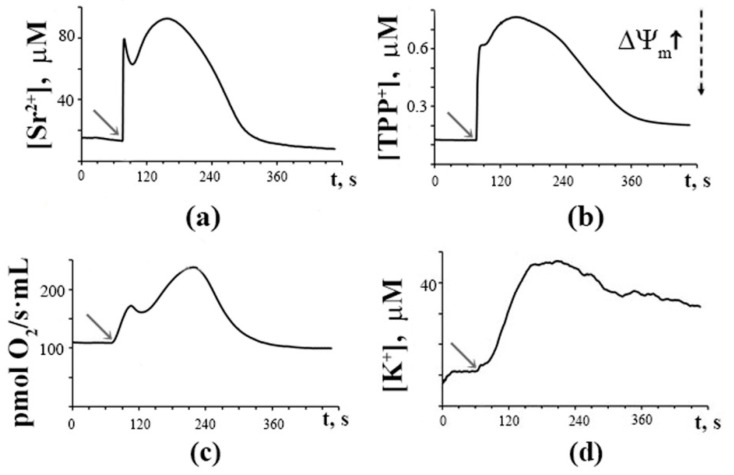
Single pulse addition of Sr^2+^ (shown by the arrow) induces rapid reversible changes in Sr^2+^ fluxes (**a**), membrane potential (**b**), respiratory rate (**c**), and slow changes in K^+^ fluxes (**d**) in the suspension of rat liver mitochondria. The incubation medium contained 20 mM sucrose, 1 mM KCl, 1 μM TPP^+^, 1 μM CsA, 1 μM rotenone, 5 mM succinic acid, and 12.5 mM Tris (pH 7.3). Addition: 45 nmol SrCl_2_/mg of mitochondrial protein. The dotted arrow indicates a change in the membrane potential. The typical traces are presented (*n* = 7).

**Figure 2 membranes-12-00667-f002:**
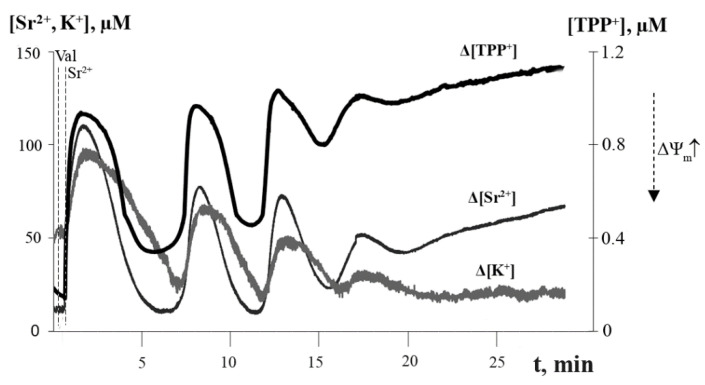
Simultaneous recording of Sr^2+^/valinomycin-induced oscillations in Sr^2+^ and K^+^ fluxes and membrane potential of rat liver mitochondria. The medium contained 20 mM sucrose, 1 mM KCl, 1.5 μM TPP^+^, 1 μM CsA, 1 μM rotenone, 5 mM succinic acid, and 12.5 mM Tris (pH 7.3). Additions: 2 ng valinomycin and 45 nmol SrCl_2_/mg of protein. The typical traces are presented (*n* = 5).

**Figure 3 membranes-12-00667-f003:**
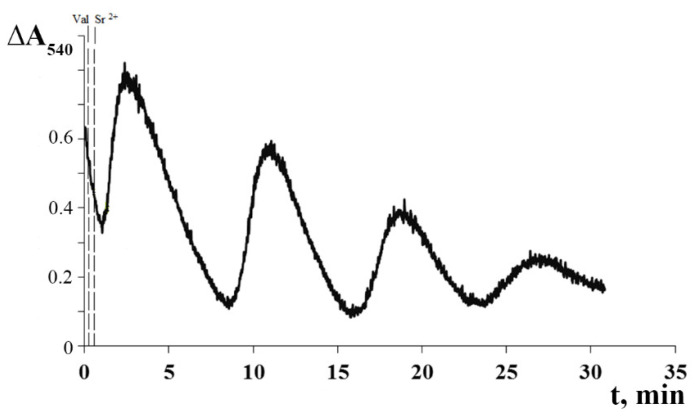
Sr^2+^/valinomycin-induced cyclic changes in matrix volume of rat liver mitochondria. The experimental conditions were the same as in Figure 2. Additions: 2 ng valinomycin and 45 nmol SrCl_2_/mg of protein. The typical traces are presented (*n* = 5).

**Figure 4 membranes-12-00667-f004:**
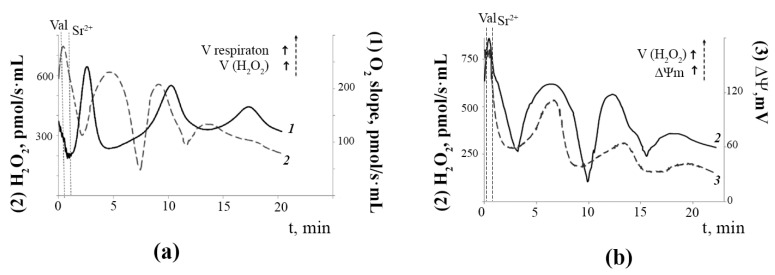
Sr^2+^/valinomycin-induced dynamic changes in the respiration rate (trace *1*), H_2_O_2_ production rate (trace *2*), and the membrane potential (trace *3*) of rat liver mitochondria: (**a**) Reciprocal changes in the rates of oxygen consumption and H_2_O_2_ production by rat liver mitochondria after the addition of Sr^2+^ and valinomycin. The conditions were the same as in Figure 2. The rate of mitochondrial respiration (O_2_ slope, (pmol/s·ml)) was measured polarographically using an Oxygraph-2k respirometer and DatLab software (Oroboros Instruments, Innsbruck, Austria). The rate of H_2_O_2_ production (H_2_O_2_, (pmol/s·ml)) by the mitochondria was determined by the Amplex Red/peroxidase assay. The typical traces are presented (*n* = 5); (**b**) Synchronous changes in the rate of H_2_O_2_ production and the membrane potential of rat liver mitochondria after the addition of Sr^2+^ ions and valinomycin. The conditions were the same as in Figure 2. The mitochondrial membrane potential was estimated with the use of tetraphenylphosphonium (TPP^+^) and an electrode selective for TPP^+^ as described in the Materials and Methods section. The amount of TPP^+^ accumulated in mitochondria was determined by measuring the difference between its initial concentration and the concentration after the addition of mitochondria. The typical traces are presented (*n* = 5).

**Figure 5 membranes-12-00667-f005:**
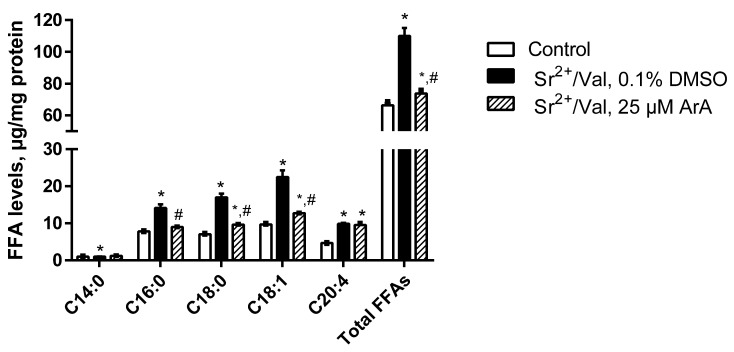
A comparative analysis of the content of the main free fatty acids in rat liver mitochondria before (control) and after the onset of Sr^2+^/valinomycin (Val)-induced ion oscillations in the absence (0.1% DMSO) or presence of 25 μM aristolochic acid (ArA), a phospholipase A_2_ (PLA_2_) inhibitor. The medium and conditions were the same as in Figure 2. The inhibitor of PLA_2_ ArA or 0.1% DMSO was added to a mitochondrial suspension 1 min before the addition of Sr^2+^. Data represent the means ± SEM of at least four independent experiments. * *p* < 0.05 vs. the control group (without additions); ^#^
*p* < 0.05 vs. the experimental group without PLA_2_ inhibitor (0.1% DMSO).

**Figure 6 membranes-12-00667-f006:**
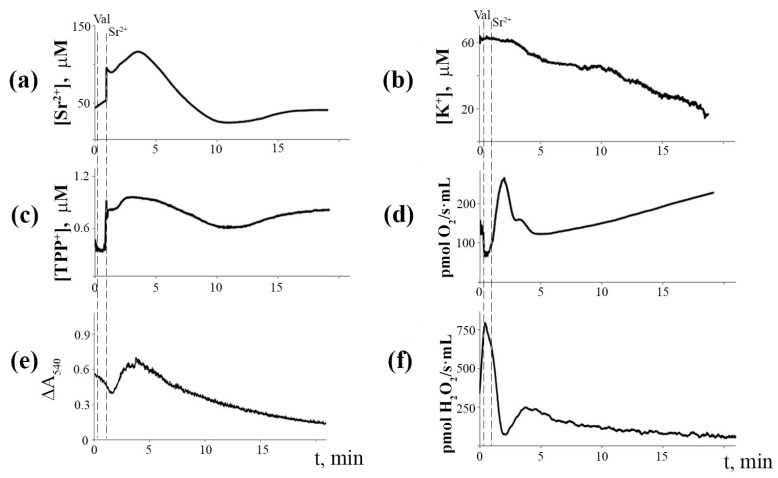
Blocking effect of aristolochic acid (25 μM), a phospholipase A_2_ inhibitor, on Sr^2+^/valinomycin-induced cyclic changes in the fluxes of Sr^2+^ (**a**), K^+^ (**b**), TPP^+^ (**c**), respiration rate (**d**), matrix volume (**e**), and H_2_O_2_ production rate (**f**) in rat liver mitochondria. The medium and conditions were the same as in Figure 2. Additions: 2 ng valinomycin and 45 nmol SrCl_2_/mg of protein. The typical traces are presented (*n* = 5).

**Figure 7 membranes-12-00667-f007:**
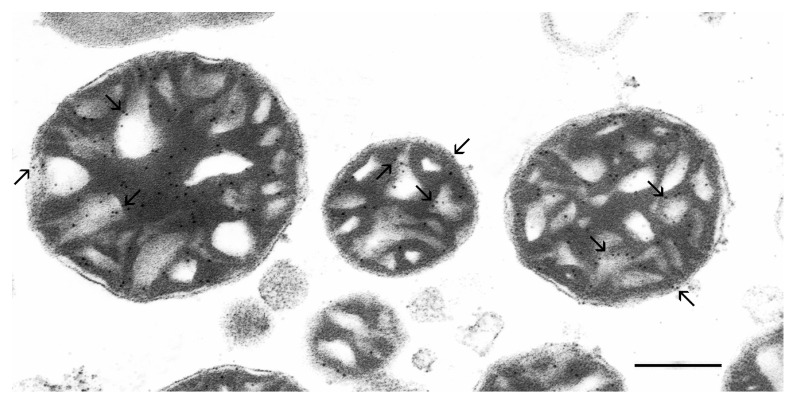
Ultrastructural localization of the group IV Ca^2+^(Sr^2+^)-dependent cytosolic phospholipase A_2_ (cPLA_2_) in isolated rat liver mitochondria. Mitochondria were incubated with specific antibodies against cPLA_2_ and secondary antibodies labeled with 10 nm colloidal gold nanoparticles. Black granules (shown by the arrows) are the binding sites for the antibodies to the target proteins. The scale bar is 0.25 μm.

**Table 1 membranes-12-00667-t001:** Phospholipase A_2_ inhibitors capable of preventing the Sr^2+^/valinomycin-induced oscillatory state of rat liver mitochondria.

PLA_2_ Inhibitor	Concentration Required to Suppress Mitochondrial Oscillations
Aristolochic acid	25 µM
Trifluoperazine dihydrochloride	10 µM
Arachidonyl trifluoromethyl ketone (AACOCF_3_)	15 µM
4-(4-Octadecylphenyl)-4-oxobutenoic acid	1 µM
Bromoenol lactone	15 µM
4′-bromophenacyl bromide	40 µM

## Data Availability

The data presented in this study are available on request from the corresponding author.

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
