# Peer review of "The Short-Term Opening of Cyclosporin A-Independent Palmitate/Sr2+-Induced Pore Can Underlie Ion Efflux in the Oscillatory Mode of Functioning of Rat Liver Mitochondria"

_membranes, 2022, doi:10.3390/membranes12070667_

Round 1
Reviewer 1 Report
There are not many related studies in this study. It can give important meaning to the continued research, but it is thought that there is little physiological or biological significance related to the disease. Therefore, it is necessary to proceed with pathological studies related to the disease.
The study started in 1976 and the most recent study was in 2015. What is the exact physiological and biological significance and meaning of the CsА-independent palmitate/Ca2+-induced permeability transition pore (PA-mPT pore)? Research trends until recently?
What, exactly, are the results more advanced compared to the 2015 study? Biological studies using confocal fluorescence microscopy are needed.
The data in Figure 1 and the data in the 2015 figure are the same. What is the significance in relation to the reproduced data?
It would have been better if the data in Figure 7 were made into data by confocal fluorescence microscopy.
Line 509-535: You have written about ROS, so I would like you to directly study it and suggest what it means.
Author Response
Please, see the attachment

Reviewer 2 Report
membranes-1786488-peer-review-v1
The aim of this work was reveal whether the PA-mPT pore formation is involved in the mechanism of ion efflux upon ion fluctuations triggered by Sr2+ in the presence of valinomycin in rat liver mitochondria.
Some minor suggestions are given below, after which the manuscript could be considered for acceptance
1- In the sections 2.1 Materials and 2.5 Determination of free fatty acids upon mitochondrial oscillations by HPLC
Please include analytical quality, company of origin and country of the following reagents or standards used
DMSO, ethanol and standard mixture of FFAs
2- Section 2.4 Estimation of the functional parameters of mitochondria
At the beginning of this section they could include a brief paragraph like the following, if the authors agree
“All assays were carried out according Protocol No. 19/2020 approved by University of Helsinki and the Russian Academy of Sciences.”
This regardless of whether it is included in the section Institutional Review Board Statement
3- Section 2.5 Determination of free fatty acids upon mitochondrial oscillations by HPLC
The levels of free fatty acids (FFAs) in rat liver mitochondria were determined by HPLC using a Pye Unicam chromatography system (Pye, England) in accordance with the conventional technique for the determination of fatty acid composition [35].
Please, after this paragraph, briefly describe the technique.
Indicate specifically type and model of chromatography instrument used, as well as corresponding acronym for the chromatographic system used (GC_MS; HPLCMS or other ) and column used.
4- If possible, include a GC or HPLC finger print of the samples as supplementary material.
Author Response
Please, see the attachment

Reviewer 3 Report
The results obtained by this study are very interesting and provide valuable insights into the understanding of the role of the mitochondria as a calcium reservoir and its implications for mitochondrial metabolism. However, some technical issues were not very clear, for example, to prevent the lipids of the tissue (liver) from uncoupling the mitochondria, it is recommended to use bovine serum albumin since free fatty acids activate the uncoupling proteins that participate in the flow calcium and also BSA allows a more precise determination of phospholipase activity. In addition, fatty acids are determined by HPLC, but according to the bibliographic reference it is by gas chromatography. Please clarify these two aspects since the interpretation of the results depends on them.
Another recommendation is that in the figures that show several strokes, a color code is used to make them more didactic for the reader and avoid confusion (for example, figure 4). Please improve the edition of the figures.
Lastly, there are some symbology errors throughout the text, especially with degrees Celsius.
Author Response
Please, see the attachment.

Reviewer 4 Report
Dear authors,
The paper you submitted presents an interesting study aiming to deeply study the mechanism of ion efflux in rat liver mitochondria.
The study is well designed and properly performed. It was proved that the efflux of ions from mitochondria down their concentration gradients can be mediated by the opening of the short-living pore induced by fatty acids and Sr2+.
This study improved the understanding of the mechanism of the oscillatory mode of mitochondrial function under both physiological and pathological conditions, improving the pieces of knowledge in this particular field.
Author Response
Response to Reviewer #4
We would like to thank the Reviewer for the comments and time in reviewing our manuscript. Enclosed, please find the revised version of our manuscript.